# Hierarchical Bias-Driven Stratification for Interpretable Causal Effect Estimation

## Abstract

Causal effect estimation from observational data is an important analytical approach for data-driven policy-making. However, due to the inherent lack of ground truth in causal inference accepting such recommendations requires transparency and explainability. To date, attempts at transparent causal effect estimation consist of applying post hoc explanation methods to black-box models, which are not interpretable. In this manuscript, we present BICauseTree: an interpretable balancing method that identifies clusters where natural experiments occur locally. Our approach builds on decision trees to reduce treatment allocation bias. As a result, we can define subpopulations presenting positivity violations and exclude them while providing a covariate-based definition of the target population we can infer from. We characterize the method's performance using synthetic and realistic datasets, explore its bias-interpretability tradeoff, and show that it is comparable with existing approaches.

## 1 Introduction

The primary task of causal inference is estimating the effect of a treatment or intervention. Evaluating the strength of a causal relationship is essential for decision-making, designing interventions, as well as evaluating the effect of a policy. As such, causal inference has high applicability across multiple fields including medicine, social sciences and policy-making.

However, the estimation of a causal effect requires the computation of "potential outcomes" i.e. the outcome an individual would experience if they had received some potential treatment, which may differ to the observed one (1). When treatment is binary, the quantity of interest is often the difference between the average potential outcomes in an all-treated scenario vs an all-untreated scenario. Estimating and evaluating causal effect from observational data is thus challenging as we only observe a single potential outcome–the one under the observed treatment–and can never observe the counterfactual outcome, lacking ground-truth labels. Furthermore, when treatment assignment is not randomized, groups that do or do not receive treatment may not be comparable in their attributes, and such attributes can influence the outcome too (i.e. confounding bias).

In addition to these fundamental challenges, in practical settings where causality is used, decision-making can often be safety-sensitive (e.g. healthcare, education). This, in turn, incentivizes "interpretable" modeling to either comply with ethics requirements or be properly communicated to interested parties. Here, *interpretability* means that each decision in the algorithm is inherently explicit and traceable, contrasting with *explainability* where decisions are justified post-hoc using an external model (2). Moreover, due to the lack of ground truth, interpretability is of greater importance in causal inference where understanding a model may be the only way to question it.

In this paper, we introduce BICauseTree: Bias-balancing Interpretable Causal Tree, an interpretable balancing method for observational data with binary treatment that can handle high-dimensional datasets. We use a binary decision tree to stratify on imblanced covariates and identify subpopulations with similar propensities to be treated, when they exist in the data. The resulting clusters act as local

naturally randomized experiments. This newly formed partition can be used for effect estimation, as well as propensity score or outcome estimation. Our method can further identify positivity violating regions of the data space, i.e., subsets of the population where treatment allocation is highly unbalanced. By doing so, we generate a transparent, covariate-based definition of the target population or "inferentiable" population i.e. the population with sufficient overlap for inferring causal effect.

Our contributions are as follows:

1. Our BICauseTree method can identify "natural experiments" i.e. subgroups with lower treatment imbalance, when they exist.
2. BICauseTree compares with existing methods for causal effect estimation in terms of bias while maintaining interpretability. Estimation error and consistency of the clusters show good robustness to subsampling in both our synthetic and realistic benchmark datasets.
3. Our method provides users with built-in prediction abstention mechanism for covariate spaces lacking common support. We show the value of defining the inferentiable population using a clinical example with matched twins data.
4. The resulting tree can further be used for propensity or outcome estimation.
5. We release open-source code with detailed documentation for implementation of our method, and reproducibility of our results.

## 2 Related work

### 2.1 Effect estimation methods

Causal inference provides a wide range of methods for effect estimation from data with unbalanced treatment allocation. There are two modelling strategies: modelling the treatment using the covariates to balance the groups, and modelling the outcome directly using the covariates and treatment assignment.

In balancing methods such as matching or weighting methods, the data is pre-processed to create subgroups with lower treatment imbalance or "natural experiments". *Matching* methods consist of clustering similar units, based on some distance metric, from the treatment and control groups to reduce imbalance. Euclidean and Mahalanobis distances are commonly used, together with nearest neighbour search. However, as the notion of distance becomes problematic in high dimensional spaces, covariate-based matching tends to become ineffective in such settings (3). *Weighting* methods aim at balancing the covariate distribution across treatment groups, with Inverse Probability Weighting (IPW) (4) being the most popular approach. Samples weights are the inverse of the estimated *propensity* scores, i.e. the probability of a unit to be assigned to its observed group. However, extreme IPW weights can also increase the estimation variance.

Contrastingly, in *adjustment* methods the causal effect is estimated from regression outcome models where both treatment and covariates act as predictors of the outcome. These regressions can be fitted through various methods like linear regression (5), neural networks (6; 7), or tree-based models (8; 9). Under this taxonomy, BICauseTree is a *balancing* method, i.e., a data-driven mechanism for achieving conditional exchangeability. Nonetheless, BICauseTree can be combined with other methods to achieve superior results. Either as propensity models in established doubly robust methods (10), or by incorporating arbitrary causal models at leaf nodes (similar to regression trees with linear models at leaf nodes (11)).

### 2.2 Positivity violations

Causal inference is only possible under the *positivity* assumption, which requires covariate distributions to overlap between treatment arms. Thus, positivity violations (also referred to as no overlap) occur when certain subgroups in a sample do not receive one of the treatments of interest or receive it too rarely (12). Overlap is essential as it guarantees data-driven outcome extrapolation across treatment groups. Having no common support means there are subjects in one group with no counterparts from the other group, and, therefore, no reliable way to pool information on their outcome had they been in the other group. Non-violating samples are thus the only ones for which we can guarantee some validity of the inferred causal effect.

There are three common ways to characterize positivity. The most common one consists in estimating propensity scores and excluding the samples associated with extreme values (also known as "trimming") (13). The threshold for propensity scores can be set arbitrarily or dynamically (14). However, since samples are excluded on the basis of their propensity scores and not their covariate values, these methods lack interpretabilty about the excluded subjects and how it may affect the target population

on which we can generalize the inference. Consequently, other methods have been developed to overcome this challenge by characterizing the propensity-based exclusion (15; 16; 17). Lastly, the third way tries to characterize the overlap from covariates and treatment assignment directly, without going through the intermediate propensity score e.g. PositiviTree (12). In PositiviTree, a decision tree classifier is fitted to predict treatment allocation. In contrast to their approach, BICause Tree implements a tailor-made optimization function where splits are chosen to maximize balancing in the resulting sub-population, whereas PositiviTree uses off-the-shelf decision trees maximizing separation. Ultimately, the above mentioned methods for positivity identification and characterization are model agnostic. In our model, BICauseTree, positivity identification *and characterization* are inherently integrated in the model, and effect estimation comes with a built-in interpretable abstention prediction mechanism.

## 2.3 Interpretability and causal inference

A predominant issue in existing effect estimation methods is their lack of interpretability. A model is considered as *interpretable* if its decisions are inherently transparent (2). Examples of interpretable models include decision trees where the decision can be recovered as a simple logical conjunction. Contrastingly, a model is said to be *explainable* when its predictions can be justified a-posteriori by examining the black-box using an additional "explanation model". Popular post-hoc explanation models include Shapley values (18) or LIME (19). However, previous works have shown that existing explainability techniques lack robustness and stability (20). Further, the explanations provided by explanation models inevitably depend on the black-box model's specification and fitness. Given that explanation models only provide unreliable justifications for black-box model decisions, a growing number of practitioners have been advocating for intrinsically interpretable predictive models (2). We further claim that causal inference, and in particular effect estimation, should be *interpretable* as it assists high-stake decisions affecting laypeople.

Causal Trees (8) are another tree-based model for causal inference that (i) leverages the inherent interpretability of decision trees, and (ii) has a custom objective function for recursively splitting the data. Although both utilize decision trees, BICauseTree and Causal Tree (CT) serve distinct purposes. BICauseTree splits are optimized for balancing treatment *allocation* while CT splits are optimized for balancing treatment *effect*, under assumed exchangeability. In other words, CT assumes exchangeability while BICauseTree "finds" exchangeability. As such, our approach is more suited for ATE estimation while CT is better suited for Conditional Average Treatment Effect estimation (8). Furthermore, in practice, causal effects are often averaged over multiple trees into a so-called Causal *Forest* (21; 22) that is no longer interpretable, and users are encouraged to use post-hoc explanation methods (23).

In addition to effect estimation, positivity violations characterization should also be interpretable for downstream users, such as policy makers. Discarding samples can hurt the external validity of any result, as there can be structural biases leading to entire subpopulation being excluded. Therefore, interpretable characterization of the overlap in a study can help policy makers better assess on whom they expect the study results to apply (15; 12). In our model, BICauseTree, we generate a covariate-based definition of the violating subpopulation. In other words, we can claim which target population our estimate of the Average Treatment Effect applies to.

## 3 BICauseTree

### 3.1 Problem setting

We consider a dataset of size $n$ where we note each individual sample $(X_i, T_i, Y_i)$ with $X_i \in \mathbb{R}^d$ is a covariate vector for sample $i$ measured prior to treatment allocation, and $T_i$ is a binary variable denoting treatment allocation. In the potential outcomes framework (24), $Y_i(1)$ is the outcome under $T_i = 1$, and $Y_i(0)$ is the analogous outcome under $T_i = 0$. Then, assuming the consistency assumption, the observed outcome is defined as $Y_i = T_i Y_i(1) + (1 - T_i) Y_i(0)$. In this paper, we focus on estimating the average treatment effect (ATE), defined as: $\text{ATE} = \mathbb{E}[Y(1) - Y(0)]$.

### 3.2 Motivation

We introduce a method for balancing observational datasets with the goal of estimating causal effect in a subpopulation with sufficient overlap. Our goals are: (i) unbiased estimation of causal effect, (ii) interpretability of both the balancing and positivity violation identification procedures, (iii) ability to handle high-dimensional datasets. Our approach utilizes the Absolute Standardized Mean Difference (ASMD) (25) frequently used for assessing potential confounding bias in observational data. Note

that our balancing procedure is entirely interpretable, although it can be used in combination with arbitrary black-box outcome models or propensity score models. Finally, our method generates a covariate-based definition of the target population on which we make inference. As such, it is tailored to sensitive domains where inference should be restricted to subpopulations with reasonable overlap.

## 3.3  Algorithm

The intuition for our algorithm is that, by partitioning the population to maximize treatment allocation heterogeneity, we may be able to find subpopulations that are natural experiments. We recursively partition the data according to the most imbalanced covariate between treatment groups. Using decision trees makes our approach transparent and non-parametric.

**Splitting criterion**  The first step of our algorithm is to split the data until some stopping criterion is met. The tree recursively splits on the covariate that maximize treatment allocation heterogeneity. To do so, we compute the Absolute Standardized Mean Difference (ASMD) for all covariates and select the covariate with the highest absolute value. The ASMD for a variable $X_j$ is defined as:

$$\text{ASMD}_j = \frac{|\mathbb{E}[X_j|T=1] - \mathbb{E}[X_j|T=0]|}{\sqrt{Var([X_j|T=1]) + Var([X_j|T=0])}}$$

The reason for choosing the feature with the highest ASMD is that it is most likely to be a confounder. Once that next splitting covariate $j_{max}$ is chosen, we want to find a split that is most associated with treatment assignment, so that we may control for the effect of counfounding. The tree finds the optimal splitting value by iterating over covariate values $x_{j_{max}}$ and taking the value associated with the lowest $p$-value according to a Fisher's exact test or a $\chi^2$ test, depending on the sample size.

**Stopping criterion**  The tree building phase stops when either: (i) the maximum ASMD is below some threshold, (ii) the minimum treatment group size falls below some threshold (iii) the total population fall below the minimum population size threshold, or (iv) a maximum tree depth is reached. All of the thresholds are user-defined hyperparameters.

**Pruning procedure**  Once the stopping criterion is met in all leaf nodes, the tree is pruned. A multiple hypothesis test correction is first applied on the $p$-values of all splits. Following this, the splits with significant $p$-values or with at least one split with significant $p$-value amongst their descendants are kept. Ultimately, given that ASMD reduction may not be monotonic, pruning an initially deeper tree allows us to check if partitioning more renders unbiased subpopulations. The implementation of the tree allows for user-defined multiple hypothesis test correction, with current experiments using Holm correction (26). The choice of the pruning and stopping criterion hyperparameters will guide the bias/variance trade-off of the tree. Deeper trees may have more power to detect treatment effect while shallower trees will be more likely to have biased effect estimation.

**Positivity violation filtering**  The final step evaluates the overlap in the resulting set of leaf nodes to identify those where inference is possible. The tree checks for treatment balance based on some user-defined overlap estimation method, with the default method being the Crump procedure (14). The positivity violating leaf nodes are tagged and then used for inference abstention mechanism, i.e. inference will be restricted to non-violating leaves.

**Estimation**  Once a tree is contracted, it can be used to estimate both counterfactual outcomes and propensity scores. For each leaf, using the units propagated to that leaf, we can model the counterfactual outcome by taking the average outcome of those units in both treatment groups. Alternatively, we can fit any arbitrary causal model (e.g., IPW or an outcome regression) to obtain the average counterfactual outcomes in that leaf. The ATE is then obtained by averaging the estimation across leaves. Similarly, we can estimate the propensity score in each leaf by taking the treatment prevalence or using any other statistical estimator (e.g., logistic regression).

**Code and implementation details**  Code for BICauseTree is released open-source, including detailed documentation under: `https://anonymous.4open.science/r/BICause-Trees-F259`. Our flexible implementation allows the user to extend the default stopping criterion as well as the multiple hypothesis correction method. BICauseTree adheres to `causallib`'s API, and can accept various outcome and propensity models.

---

**Algorithm 1 BICauseTree**

---

**Inputs:** root node $N_0$, $X$, $T$, $Y$
Call *Build subtree*($N_0$, $X$, $T$, $Y$)
Do multiple hypothesis test correction on all split $p$-values
**Pruning procedure**: keep splits with either (i) a significant $p$-value or (ii) at least one descendant with a significant $p$-value
Mark leaf nodes that violate positivity violation criterion

---

---

**Algorithm 2 Build subtree**

---

**Inputs:** current node $N$, $X$, $T$, $Y$
**if** Stopping criteria not met **then**
    Find and record in $N$ the covariate with maximum ASMD: $maxASMD := max_i(ASMD_i)$
    Find and record in $N$ the split value with the lowest $p$-value according to a Fisher test/$\chi^2$ test
    Record the $p$-value for this split in $N$
    Split the data $X, T, Y$ into $X_{left}, T_{left}, Y_{left}$ and $X_{right}, T_{right}, Y_{right}$ according to $N$'s splitting covariate and value
    Add two child nodes to $N$: $N_{left}$ and $N_{right}$
    Call *Build subtree*($N_{left}, X_{left}, T_{left}, Y_{left}$)
    Call *Build subtree*($N_{right}, X_{right}, T_{right}, Y_{right}$)
**end if**

---

## 4 Experiment and results

### 4.1 Experimental settings

In all experiments–unless stated otherwise–the data was split into a training and testing set with a 50/50 ratio. The training set was used for the construction of the tree and for fitting the outcome models in leaf nodes, if relevant. Causal effects are estimated by taking a weighted average of the local treatment effects in each subpopulation. At the testing phase, the data is propagated through the tree, and potential outcomes are evaluated using the previously fitted leaf outcome model. We performed 50 random train-test splits, which we will refer to as *subsamples* to avoid confusion with the tree partitions. For each subsample, effects are only computed on the non-violating samples of the population. In order to maintain a fair comparison, these samples are also excluded from effect estimation with other models and with ground truth. All results are shown after filtering positivity-violating samples.

**Baseline comparisons** We compare our method to doubleMahalanobis Matching, Inverse Probability Weighting (IPW), and Causal Tree (CT). In Mahalanobis Matching (27; 28), the nearest neighbor search operates on the Mahalanobis distance: $d(X_i, X_j) = (X_i - X_j)^T \Sigma^{-1}(X_i - X_j)$, where $\Sigma$ is alternatively the estimated covariance matrix of the control and treatment group dataset. In Inverse Probability Weighting (4), a propensity score model estimates the individual probability of treatment conditional on the covariates. The data is then weighted by the inverse propensities $P(T = t_i \mid X = \boldsymbol{x}_i)^{-1}$ to generate a balanced pseudo-population. In Causal Tree (8), the splitting criterion optimizes for treatment *effect* heterogeneity (see section 3.1 for further details). We use a Causal Tree and not a Causal Forest to compare to an estimator which is equally interpretable as our estimator. We also compare our results to an unadjusted marginal outcome estimator, which will act as our "dummy" baseline model. As using a single Causal Tree for our interpretability goal gives rise to high estimation bias, Causal Tree was excluded from the main manuscript for scaling purposes. We refer the reader to sections A.5, A.6 and A.7 for a comparison with CT. For synthetic experiments, we use the simplest version of our tree which we term BICauseTree(Marginal) where the effect is estimated from taking average outcomes in leaf nodes. For real-world experiments, we compare BICauseTree(Marginal) with BICauseTree(IPW), an augmented version in which an IPW model is fitted in each leaf node. To compare estimation methods, we compute the difference between the estimated ATE and the true ATE for each subsample (or train-test partition) and display the resulting distribution of estimation biases in a box plot. Further experimental details, including hyperparameters, can be found in the Appendix under section A.9.

### 4.2 Synthetic datasets

We first evaluate the performance of our approach on two synthetic datasets. We first demonstrate BICauseTree's ability to identify subgroups with lower treatment imbalance on a dataset which we will refer to as the "natural experiment dataset" in the following. We further exemplify BICauseTree's identification of positivity violating samples on a dataset we refer to as the "positivity violations dataset". Due to the interaction-based nature of the data generation procedure, we additionally compare our approach to an IPW estimator with a Gradient Boosting classifier propensity model, referred to as IPW (GBT) in both synthetic experiments. This choice ensures a fair comparison across estimators.

**Identifying natural experiments** For the natural experiment dataset, we considered a Death outcome $D$, a binary treatment of interest $T$ and two covariates: Sex $S$ and Age $A$. We defined four sub-populations, where each constituted a natural experiment with a truncated normal propensity distribution centered around a pre-defined constant value and variance (see details in Section A.4.1). Then, individual treatment propensities were sampled from the corresponding distribution and observed treatment values were sampled from a Bernoulli distribution parameterized with the individual propensities. No positivity violation was modeled in this experiment. Ultimately, $X = (S, A)$ is the vector of covariate values in $\mathbb{R}^2$ with the sample size chosen as $n = 20,000$. The marginal distribution of covariates follows: $S \sim \text{Ber}(0.5)$ and $A \sim \mathcal{N}\left(\mu, \sigma^2\right)$ where $\mu = 50$ and $\sigma = 20$. Figure A1 in A.5.1 shows the partition obtained from training BICauseTree on the entire dataset. Our tree successfully identifies the subpopulations in which a natural experiment was simulated. Figure 1a shows the estimation bias across subsamples. In addition to being transparent, BICauseTree has lower bias in causal effect estimation compared to all other methods, excluding IPW(GBT) which has comparable performance. Despite its higher estimation variance, Matching has low bias, probably due to covariate space being well-posed and low-dimensional. Contrastingly, the logistic regression in IPW(LR) is not able to model treatment allocation as the true propensities are generated from a noisy piecewise constant function of the covariates resulting in a threshold effect that explains its poor performance. The non-parametric, local nature of both Matching and BICauseTree thus contrasts with the parametric estimation by IPW(LR). Further results on the BICauseTree's calibration and covariate partition can be found in the Appendix, under section A.5.1.

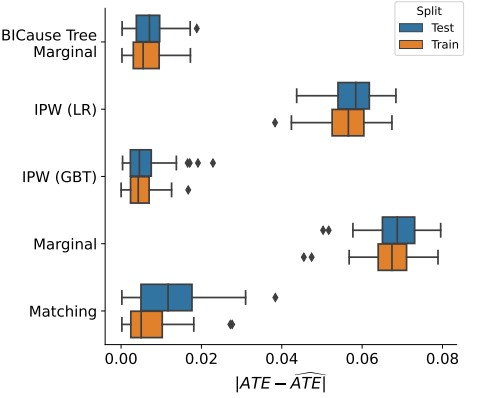
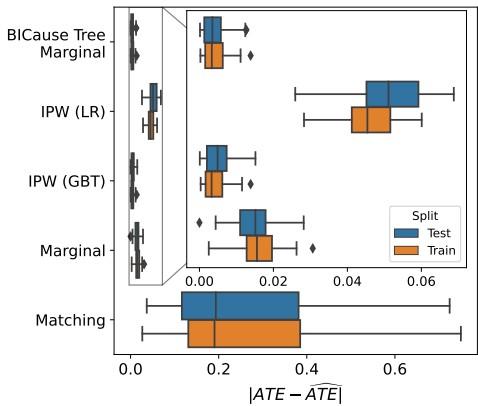

(a) Estimation bias for the natural experiment dataset (see subsection 4.2) across 50 subsamples, with $N = 20,000$

(b) Estimation bias for the positivity violations dataset (see subsection 4.2) across 50 subsamples, after excluding positivity violating leaf nodes with $N = 20,000$.

Figure 1: Results on the synthetic datasets

**Identifying positivity violations** For the positivity violations dataset, we consider a synthetic dataset with a Death outcome $D$, a binary treatment of interest $T$, and three Bernoulli covariates –Sex $S$, cancer $C$ and arrhythmia $A$– such that $X = (S, C, A)$ (see Section A.4.2 for further details). As for the natural experiment dataset, we modeled treatment allocation with stochasticity by sampling propensities from a truncated gaussian distribution first. Treatment allocation was simulated to ensure that overlap is very limited in two subpopulations: females with no cancer and no arrhythmia are rarely treated, while males with cancer and arrhythmia are almost always treated. Figure A2 in A.5.2 shows the partition obtained from training BICauseTree on the entire dataset, confirming that

268  BICauseTree excludes the subgroups where positivity violations were modeled. On average, 67.1%
269  of the cohort remained after positivity filtering with very little variability across subsamples. Thanks
270  to the interpretable nature of our method, we are able to identify these subgroups as a region of
271  the covariate space. As seen in Figure 1b, after filtering violating samples the effect estimation by
272  BICauseTree remains unbiased and with low variance. Our estimator compares with IPW(GBT)
273  while being interpretable. The IPW(LR) estimator is more biased than BICauseTree. This may be
274  due to the extreme weights in the initial overall cohort. In spite of filtering samples from regions
275  with lack of overlap–as defined by BICauseTree–the remaining propensity weights may be biased,
276  which would ultimately induce a biased effect estimation. Estimation variance is comparable across
277  methods, except for Matching which is both more biased and has higher variance than all other
278  estimators. Further results on the BICauseTree's calibration and covariate partition can be found in
279  the Appendix, under section A.5.2.

### 4.3   Realistic datasets

281  **Causal benchmark datasets**   We use two causal benchmark datasets to show the value of our
282  approach. The twins dataset illustrates the high applicability of our procedure to clinical settings. It
283  is based on real-world records of $N = 11,984$ pairs of same-sex twin births, and has 75 covariates.
284  It tests the effect of being born the heavier twin (i.e. the treatment) on death within one year (i.e.
285  the outcome), with the outcomes of the twins serving as the two potential outcomes. We use the
286  dataset generated by *Neal et. al* (29), that simulates an observational study from the initial data by
287  selectively hiding one of the twins with a generative approach. We also ran our analysis on the *2016*
288  *Atlantic Causal Inference Conference* (ACIC) semisynthetic dataset with simulated outcomes (30).
289  For ACIC, given that trees are data greedy, and due to the smaller sample size ($N = 4,802$) relative
290  to the number of covariates ($d = 79$), the models were trained on 70% of the dataset.

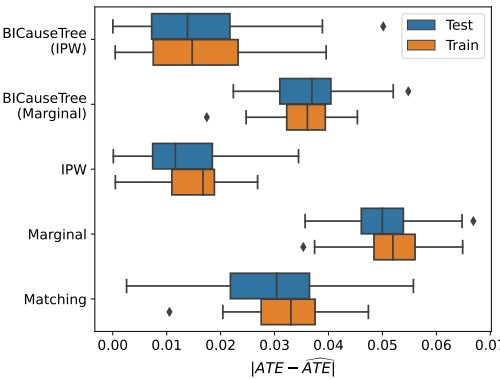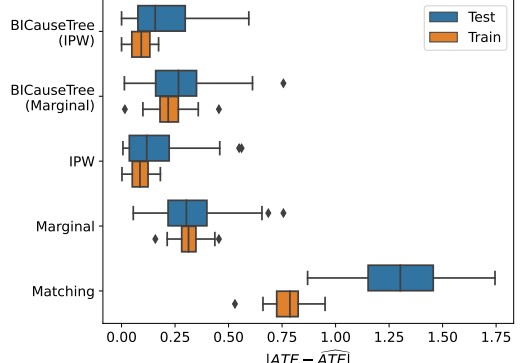

Figure 2: Estimation bias for the twins dataset ($N = 11,984$) across 50 subsamples, excluding positivity violating leaf nodes.

Figure 3: Estimation bias for the ACIC dataset ($N = 4,802$) across 50 subsamples, excluding positivity violating leaf nodes.

291  **Effect estimation**   Figure 2 shows the distribution of the estimation biases across subsamples on
292  the twins dataset, comparing to the baseline models. Here, our BICauseTree(Marginal) estimator
293  is less biased than the marginal estimator. Augmenting our tree with an IPW outcome model –
294  BICauseTree(IPW) – further decreases estimation bias, making it comparable with IPW, both w.r.t
295  bias and estimation variance. Figure 3 compares the estimation biases across estimators on the ACIC
296  dataset. Here, both BICauseTree models compare with IPW in terms of bias and estimation variance.
297  **Bias-interpretability tradeoff**   We expect a bias-interpretability tradeoff, where deeper trees are
298  less biased but more complex to understand, while shallower trees are less accurate but easier
299  to comprehend. Figure 4 shows how estimation bias in leaf nodes decreases as we increase the
300  maximum depth hyperparameter of our BICauseTree(Marginal) in the twins dataset. Here, each
301  circle in the plot represents a leaf node, and the dotted line shows the average bias with an IPW
302  estimator. The shaded area represents the 95% confidence interval (CI) for IPW. As seen in the
303  plot, there is some overlap between the 95% CI for IPW and the estimation bias of deeper trees.
304  The remaining gap thus represents the need for a more complex outcome model in the leaves, or in
305  other words the estimation bias that was traded against interpretability here. Similarly, in figures 2
306  and 3 we notice how augmenting our partition with an IPW leaf outcome models has decreased the

307 estimation bias at the cost of transparency. Ultimately, figure 4 shows that bias reduction is consistent
308 beyond a maximum depth parameter of 5. The robustness of our estimator w.r.t the maximum depth
309 hyperparameter is likely due to our statistical pruning procedure. A similar figure is shown in Section
310 A.7 for the ACIC dataset.

311 **Interpretable positivity violations filtering**
312 As previously discussed, BICauseTree provides
313 a built-in method for identifying positivity vi-
314 olations in the covariate space directly. After
315 positivity filtering, effect was computed on an
316 average of $99.5\%$ ($\sigma = 0.006$) of the population
317 on the twins dataset, and an average of $85.9\%$
318 ($\sigma = 0.093$) of the ACIC dataset.
319 Figure A4 in the Appendix shows the tree parti-
320 tion for the twins dataset. One leaf node was de-
321 tected as having positivity violations ($N = 106$).
322 The twins example illustrates the real-world im-
323 pact of having a covariate-based definition of the
324 non-violating subpopulation. Here, we are able
325 to claim that our estimate of the effect of being
326 born heavier might not be valid for newborns
327 that fit the criteria for this specific violating node.
328 This capability of BICauseTree is highly valu-
329 able in any safety-sensitive setting. Consider a
330 scenario where the "at-risk" twin benefits from

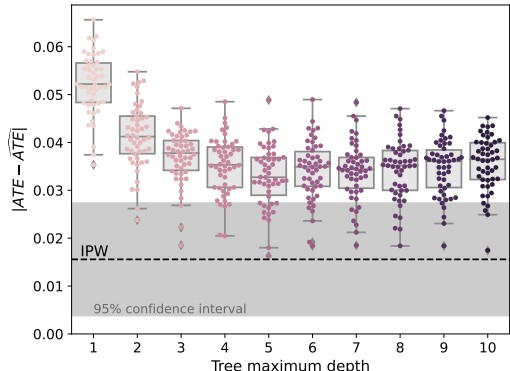

Figure 4: Estimation bias when comparing *BICauseTree(Marginal)* with varying maximum depth parameters with the average bias of IPW (dotted), on the twins training set ($N = 5,992$).

331 a follow-up visit after birth, and that the true effect of the intervention is higher in the positivity
332 violating subpopulation. Extrapolating the estimated effect of the exposure to the entire cohort may
333 be dangerous to the infants in this subgroup. It is thus essential for practitioners to know which
334 population the inferred effect applies to, which would not have been possible using alternative
335 non-interpretable methods for identifying positivity violations e.g. IPW with weight trimming, as they
336 provide an opaque exclusion criterion. Additionally, note that the positivity violation identification
337 remains transparent regardless of the chosen propensity or outcome model at the leaves.

338 **Propensity score estimation** Alternative use-cases for BICauseTree include using the partition as a
339 propensity model. Given the importance of calibrated propensity scores (31), Figure 5 compares the
340 calibration of the propensity score estimation of BICauseTree with the one from logistic regression
341 (IPW) on the testing set of the twins dataset. As expected, logsitic regression, which has better data
342 efficiency, has better, less-noisy calibration. However, BICauseTree still shows satisfying calibration
343 on average. Section A.6 in the Appendix shows the calibration plots for the estimation of potential
344 outcomes on the twins dataset. Section A.7 shows calibration plots for the ACIC dataset.

345 **Tree consistency** To evaluate the consistency
346 of our clustering across subsamples, we train
347 our tree on 70% of the dataset and compute the
348 adjusted Rand index (32) (see further details in
349 section A.2). We chose not to train on 50% of
350 the data here as most of the inconsistency would
351 then be due to the variance between subsamples.
352 For the twins dataset, the Rand index across
353 50 subsamples of sample sizes $N = 8,388$,
354 is equal to $0.633$ ($\sigma = 0.208$). For the ACIC
355 dataset, the Rand index across 50 subsamples
356 of sample sizes $N = 3,361$, is equal to $0.314$
357 ($\sigma = 0.210$) which shows that our tree is not
358 consistent across subsamples if sample size is

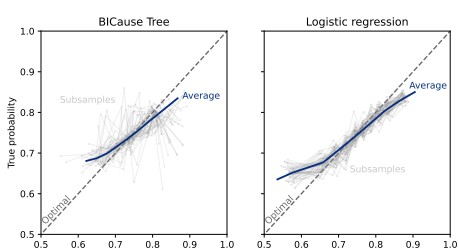

Figure 5: Calibration of the propensity score estimation for the twins dataset

359 not substantial. However, we exemplify consistent identification of the positivity population, with
360 the variance of the percentage of positive samples equal to $\sigma = 0.006$ and $\sigma = 0.093$ (see paragraph
361 4.3) in the twins and ACIC dataset respectively. Ultimately, throughout our experiments, we noticed
362 how consistency starts to decrease if the maximum depth hyperparameter increases past a certain
363 threshold. As a heuristic, we would recommend users to test tree consistency across subsamples
364 when tuning this hyperparameter.

# 5   Discussion

**Strengths and limitations of our approach**  Following our discussion on the bias-interpretability tradeoff, we acknowledge that in complex data settings where finding sub-populations that enclose natural experiments is difficult, the resulting BICauseTree partition may have remaining bias in some leaf nodes, and ultimately render some estimation bias. This bias is, however, traded-off with enhanced interpretability, as previously discussed. Nonetheless, as exemplified in this work, the performances of BICauseTree remains comparable, with estimation bias being only slightly larger than common models such as IPW. We further emphasize the fact that its strength resides in the combination of (i) the performance of the estimator with (ii) the interpretability of the balancing and positivity identification procedures, and (iii) the ability to handle high-dimensional datasets. Another advantage of BICauseTree is its ability to identify complex interaction features that are significantly correlated to treatment allocation. Indeed, in leaf nodes that come directly from a significant split, the root-to-leaf path is an interaction significantly associated with treatment allocation after multiple hypothesis test correction. Common alternatives to identify such interactions include exhaustive enumeration of all pairs of feature interactions, or complex feature engineering (33). However these approaches either lack transparency or become problematic in high-dimensional datasets. Furthermore, the tree nature of our approach is a major strength. BICauseTree is a non-parametric estimator that inherit the desirable empirical properties of regression forests—such as stability, ease of use, and flexible adaptation to different functional forms. Finally, the computational expense induced from fitting a BICauseTree is manageable: it is roughly comparable to IPW and CausalForest, and substantially lower than for Matching (see detailed compute times in Section A.9) Our work has the following limitations: (i) due to its tree structure, BICauseTree has lower data efficiency than most other estimators, including IPW. However the data efficiency of BICauseTree was superior to that of CT in our experiments. (ii) our tree design has some lunging dependence on sample size. While our estimation of ASMD is independent of sample size, the variance of our estimator, $\widehat{ASMD}$, is dependent on $n$. Furthermore, having chosen the splitting covariate, the choice of a split point is biased towards equal split subgroups. (iii) our individual splitting decisions do not consider interactions and instead only consider the marginal association of covariates with treatment.

**Applicability of BICauseTree**  We claim that BICauseTree is highly relevant when causality is examined in a context with substantial safety and ethical concerns. We consider the transparency of our built-in approach to positivity violation identification particularly relevant to fields such as epidemiology, econometrics, medicine, and policy-making. The social impact of our work, and its relevance to the upcoming policies for Artificial Intelligence is further discussed in section A.10. In addition, we claim that our ability to identify violating regions of the covariate space is key for experimental design. Fitting a BICauseTree to an existing dataset will advise practitioners on which individuals we currently lack data to infer an effect on, which will in turn inform them on the specific subpopulations they need to recruit from, in a potential next study.

**Conclusion and future work**  Here, we introduced a model able to detect positivity violations directly in the covariate space, perform effect estimation comparable to existing methods, while allowing for interpretability. We demonstrated our model's performance on both synthetic and realistic data, and showcased its usefulness in the principle challenges of causal inference. Future work may include extension to a non-binary tree, where we allow splitting to more than two nodes. This could be done for instance by fitting a piece-wise constant function that predicts treatment and finds the potentially multiple thresholds for optimized hetereogenous subgroups. In addition, to refine our pruning procedure, we can account for the intrisic ordering of the $p$-values of the splits using sequential multiple hypothesis testing (34; 35; 36). Furthermore, following the work of (8) on the "honest effect" in Causal Forests, we may use a subset of the data for fitting the partition of the tree and another distinct subset for fitting the outcome or propensity models in each leaf node. This procedure however requires having many samples. Another alternative to current model fitting, which is done independently in each leaf, is to partially pool estimates across the clusters and fit a multilevel outcome model with varying intercepts or varying slopes for treatment coefficients (37). In terms of estimation, one may investigate the performance of bagging multiple BICauseTrees into a BICauseForest, similarly to Causal Forest. Aggregating trees would however defeat the interpretability purpose. Finally, similarly to positivity-violating nodes, future work may explore the possibility of excluding leaf nodes with high maximum ASMD, under the premises that these subgroups do not enclose natural experiments.

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
