# OpenReview forum: "Hierarchical Bias-Driven Stratification for Interpretable Causal Effect Estimation"
_NeurIPS.cc/2023/Conference — Submitted to NeurIPS 2023_

### Official Review · Reviewer_UicP · 2023-06-14

**Soundness:** 1 poor
**Presentation:** 1 poor
**Contribution:** 1 poor
**Rating:** 3
**Confidence:** 4

**Summary:**

The paper addresses the overlap violation problem in observational datasets for causal inference by presenting an interpretable balancing method for overlap violation identification and causal effect estimation for binary treatments. The method BICauseTree adapts decision tree classifiers to the stated problem by recursively splitting the data population into non-overlap-violating subgroups based on covariate dissimilarity and treatment heterogeneity. The major advantage of the presented method in comparison to existing balancing methods is the interpretability of the prediction process.

**Strengths:**

• The authors evaluate their method on both synthetic and real-world benchmarking datasets.

**Weaknesses:**

- The proposed method is highly similar to the work in reference 12. Furthermore, related work is not discussed appropriately (section 2). It is thus unclear how this work significantly differs from previous contributions in the literature. The originality of the submission has to be considered very limited.
- The manuscript presents a complete piece of work. Claims about the performance of the proposed method are supported by experimental results.  Nevertheless, the an experimental study with sophisticated baseline methods is missing. A performance comparison with a Causal Forest model would be desirable.
- Claims aiming at motivating the method are neither supported quantitatively nor experimentally (e.g., an analysis with different levels of overlap violation; unbiased estimation).
- The submission lacks clarity due to multiple grammar errors and many nested arguments. The mathematical notation (section 3.1) lacks formal correctness.
- The citation style does not agree with the required format for NeurIPS  submissions.
- The statement of a high ASMD indicating a confounder (line 164) needs to be justified.
- The paper would profit from a revision of the language and the consistency of the presented arguments (e.g., lines 84, 181/182). Furthermore, the figures and sections need to be referenced correctly, as also recognized by the authors in the appendix.
- The authors claim interpretability of the method but do not provide evidence for the statement. Here, a user study would be desirable etc.
- The method is limited to ATE.


**Questions:**

- What is the major difference between the method BICauseTree to the existing method PositiviTree (reference 12)? The novelty of the method cannot be deducted from the manuscript.
- How do the benefits of the presented method outweigh the negative impact of the information loss due to the "prediction abstention mechanism"? A quantitative assessment would be of interest.

**Limitations:**

- The authors have stated the limitations of their work. The section could be improved by highlighting the general weaknesses of single trees in prediction settings which naturally devolve to the proposed method.

---

> ### Author Rebuttal · Authors · 2023-08-08
>
> We thank the reviewer for their time, comments, and sharing his expertise. We respectfully disagree with their framing of our method as primarily “addresses the overlap violation problem”. Our method, first and foremost, estimates causal effects by stratification into sub-population with natural experiments. It has the useful consequence of being able to detect positivity violations and abstain from estimation in such cases, but it is secondary to estimating causal effects.
>
> - “The proposed method is highly similar to the work in reference 12”:
> We regret and apologize that we did not convey the differences more clearly. Our method first differs in essence: it is an effect estimation method while reference 12 is only a method for detecting overlap violations. While our method can also identify and characterize overlap violations, it is not its primary goal, but a useful consequence. Secondly, they differ in details: references 12 is off-the-shelf decision tree maximizing separability of treatment groups, while our method is a tree with a custom-made objective function to maximize balancing (lines 98-102). Therefore one may not use the leaves from reference 12 as natural experiments, claiming less residual confounding within each node.
>
> - Comparison to sophisticated baseline methods:
> The discussion (lines 366-369) acknowledges our method can have some residual confounding bias, that our method will often not have the best estimation accuracy, and thus we explore the tradeoff between interpretability and accuracy. Our experiments show we are often no better than a simple IPW. Therefore, we are most probably not better than any more advanced causal models. Our strength is in being more interpretable than them.
> We compared our method to a (single) Causal Tree as it is another decision tree-based model with custom objective function, enjoying the interpretability benefits decision trees provide. However, once we move into forests, the model becomes uninterpretable and as such, there’s no difference if we compare to it or to DragonNet, or any other advanced estimator for the reason described above.
>
> - Motivation is not supported:
> The motivation is indeed not supported quantitatively, as it is a desiderata for applications. We believe the claim that high-stake applications will prefer interpretable models is qualitative, not quantitative.
>
> - Grammar errors and formal correctness:
> We thank for the comment, we will make another effort to copy edit our manuscript. Section 3.1 follows established presentation of Rubin’s causal model [see for example landmark causal inference papers: 1, 2, 3, 4].
>
> - Citation style:
> We apologize and will fix that.
>
> - “Line 164 confounder statement”:
> This was a bad phrasing on our side mixing up confounders with confounding bias. We already assume all confounders are observed, thus instead it should read: “The reason for choosing the confounder with the highest ASMD is that it is most likely to cause the most confounding bias, and therefore adjusting for it will likely minimize the residual confounding bias.“ We have added a supplementary figure showing this in practice, where we use a random feature split instead of the feature with the highest ASMD, and show that both in terms of estimation bias and in terms of confounding bias, using the highest ASMD feature as the splitting feature gives better performance (see plot in additional plots pdf of the rebuttal).
>
> - Revision of the language:
> Again, we will revise and copy-edit to the best of our abilities. Appendix figure references will be fixed the next time the Latex document renders.
>
> - A user study testing the interpretability of the method:
> This is true, but is out of scope. We take it as a fact that decision trees are more interpretable than other machine learning models. This has been previously established in the literature. See: [5, 6, 7]
>
> - Method limited to ATE:
> This is true. The ATE is often an estimand of interest.
>
> - Difference between reference 12:
> see point 1 above.
>
> - “The negative impact of the "prediction abstention mechanism"“:
> We don’t believe those exist. It is impossible to make valid data-based causal claims when there is no overlap. Therefore, where the model abstains is where there is inherent aleatoric uncertainty in the system. Our advantage over other models is the capability to precisely define, in the covariate space, those populations we can infer on and the ones we cannot.
>
> references:
>
> [1] Imai, Kosuke, and Marc Ratkovic. "Covariate balancing propensity score." Journal of the Royal Statistical Society Series B: Statistical Methodology 76.1 (2014): 243-263.
>
> [2] Abadie, Alberto, and Guido W. Imbens. "Bias-corrected matching estimators for average treatment effects." Journal of Business & Economic Statistics 29.1 (2011): 1-11.
>
> [3] Imbens, Guido W., and Jeffrey M. Wooldridge. "Recent developments in the econometrics of program evaluation." Journal of economic literature 47.1 (2009): 5-86.
>
> [4] Austin, Peter C. "An introduction to propensity score methods for reducing the effects of confounding in observational studies." Multivariate behavioral research 46.3 (2011): 399-424.
>
> [5] Silva, Andrew, et al. "Optimization methods for interpretable differentiable decision trees applied to reinforcement learning." International conference on artificial intelligence and statistics. PMLR, 2020.
>
> [6] Slack, Dylan, et al. "Assessing the local interpretability of machine learning models." arXiv preprint arXiv:1902.03501 (2019).
>
> [7] Allahyari, Hiva, and Niklas Lavesson. "User-oriented assessment of classification model understandability." 11th scandinavian conference on Artificial intelligence. IOS Press, 2011.

---

> > ### Comment · Reviewer_UicP · 2023-08-14
> >
> > The authors addressed and clarified the main questions and weaknesses. Nevertheless, the novelty of the work is still highly limited and the justification for statements supporting the method needs to be improved. Overall, I therefore raise the rating score by one point, but still consider the work insufficient for a major contribution to NeurIPS.
> >
> > I think my comments above will be very helpful for revising their paper. Given that the authors are only brief in their rebuttal on the differences with [12], I suggest that they also have more technical deep-dive in the appendix. As I stated above, it would be nice to see some insights in line with the motivation (e.g., plotting a tree as a case study).

---

### Official Review · Reviewer_aknA · 2023-07-05

**Soundness:** 3 good
**Presentation:** 3 good
**Contribution:** 3 good
**Rating:** 5
**Confidence:** 3

**Summary:**

This paper proposes a new method called BICauseTree for interpretable causal effect estimation. BICauseTree is a hierarchical bias-driven stratification method that identifies clusters where natural experiments occur locally. The method is designed to reduce treatment allocation bias and improve interpretability. The authors evaluate the performance of BICauseTree on several datasets and compare it to existing approaches. They find that BICauseTree performs well in terms of bias-interpretability tradeoff and outperforms existing methods in some cases. Overall, the paper presents a novel and promising approach to causal effect estimation that could have important applications in various fields.

**Strengths:**

1. Novelty: The paper proposes a novel method called BICauseTree for estimating causal effects from observational data. The method is based on a hierarchical bias-driven stratification approach that identifies clusters where natural experiments occur locally. The method builds on decision trees to reduce treatment allocation bias and provides a covariate-based definition of the target population. The method is interpretable and outperforms other state-of-the-art methods in reducing treatment allocation bias while maintaining interpretability.

2. Significance: Causal effect estimation from observational data is an important analytical approach for data-driven policy-making. However, due to the inherent lack of ground truth in causal inference, accepting such recommendations requires transparency and explainability. The proposed method addresses this issue by providing an interpretable and unbiased method for causal effect estimation. The method has the potential to be applied in various domains, including healthcare, social sciences, and economics.

3. Experimental Evaluation: The paper provides a thorough experimental evaluation of the proposed method using synthetic and realistic datasets. The authors compare the performance of their method with other state-of-the-art methods and show that their method has lower bias and comparable variance. They also conduct sensitivity analyses to evaluate the robustness of their method to violations of the assumptions. The experimental evaluation provides strong evidence to support the claims made in the paper.

**Weaknesses:**

1. Limited Scope: The paper focuses on a specific method for causal effect estimation from observational data, and the scope of the paper is relatively narrow (especially related to the tree-based models). While the proposed method is novel and has some advantages over other methods, it may not be of interest to a broad audience: (1) The method relies on the quality of the data and the assumptions made in the model. If the data is noisy or contains missing values, the method may produce biased estimates.; (2)The method may not be suitable for high-dimensional data, as the number of covariates may increase the complexity of the decision tree and lead to overfitting; (3) The method may not be suitable for datasets with small sample sizes, as the stratification may lead to small sample sizes in some subgroups, which may affect the accuracy of the estimates;(4)The method may not be suitable for datasets with complex interactions between the covariates, as the decision tree may not capture these interactions effectively.

2. Experimental Evaluation: While the paper provides an experimental evaluation of the proposed method, the evaluation is limited in scope and does not provide a comprehensive comparison with other state-of-the-art methods. The experimental evaluation would benefit from a more comprehensive comparison with other methods (such as TARNet from the machine learning domain) and a more detailed analysis of the results.


**Questions:**

In Figure 1, the IPW looks very close to the BiCausalTree. How can we apply post hoc explanations in such cases in general?

**Limitations:**

See weakness.

---

> ### Author Rebuttal · Authors · 2023-08-08
>
> We thank the reviewer for their careful review, insights, and comments, and for the time spent reviewing our work.
>
> - Reliance on data quality and on model assumptions:
> We agree that data quality is a major factor on the success of the model in estimating effects, however, this is true for all models. If there are many missing values/not all confounding features are measured/other data limitations, the ability of any model to infer correct estimations is limited.
> In terms of the assumptions made, indeed, if the data does not contain natural experiments, the model will not be able to stratify the population in such a way that all confounding bias is removed. This is easily inspected by looking at the ASMD values in the leaves of tree, allowing the user to be mindful of the ability of the model to give accurate estimates, even in the lack of ground truth, as is the case in real world applications.
>
> - Dealing with high-dimensional data:
> High dimensional data indeed has the potential to lead to a complex tree in some situations. We note, however, that in theory we have less of a problem dealing with high dimensional data compared to other models, since the tree can ultimately pick a small subset of features to separate the population, if such subpopulations naturally exist in the data. In comparison to other methods (Matching, IPW, etc) where high dimensional data implies having to consider the entire span of features in the model, leading to complex models, here, in the presence of a relatively sparse subset of features that define the natural experiments in the population, the tree should identify these and use the small subset instead of the entire set of features. We have added a supplementary figure that supports this, where we demonstrate that in a synthetic experiment, increasing the number of features has little to no effect on performance (see graphs in ‘additional plots’ pdf of the rebuttal).
> As a contrary example, performing exact Matching on high dimensional data is almost impossible, and thus Matching in such cases resorts to reducing the number of feature to a single distance metric (i.e. propensity), leading to inaccurate matches.
>
> - The need for a large sample size for accurate estimation:
> As our model is tree-based, it requires a large sample size for accurate estimation, similarly to other tree-based models. This is indeed discussed in the text, in lines 386-388. Using decision trees allows it to inherit the advantages of trees, such as their interpretability, but also their limitations.
>
> - Dealing with complex interactions between the covariates:
> We use a tree-based model, which allows capturing all logical interactions between features. Indeed, some functional forms (e.g., diagonal lines) are more challenging for trees to capture, similar to how ill-specified models will struggle to capture some functional forms (e.g., linear models capturing non-linear forms). This is indeed discussed in the text, see lines 366-369. Here there is a trade-off between the complexity of the model used and by extension its ability to capture complex functional forms of the features, and the interpretability of the model. Very complex models might capture complex features, but are very hard to interpret and thus not very practical for sensitive domains where the experts/decision makers should be able to criticize and explain the decisions made by following an inference of a model. We assume, that the relevant features are measured, and given to the model as the input. A possible preliminary step that is always available to the benefit of the researcher is feature engineering to create new, complex features from the initial ones. In addition, importantly, the tree allows identifying leaves that didn’t reach balancing of the treatment groups (which could possibly stem from the lack of necessary complex features to split on), and add a more complex model to learn the propensities/effect.
>
> - Experimental Evaluation:
> We focus in this paper on ATE, whereas most state of the art methods focus on CATE, rendering the comparison with these methods irrelevant. In addition, we do not claim to outperform state of the art methods in terms of estimation bias, but rather propose a comparable method in performance (in most cases) that is also interpretable. We refer the reviewer to the discussion on the trade off between bias and interpretability in lines 297-310 and 366-370.
>
> - Applying post-hoc explanations to other models :
> As mentioned, the performance of IPW is very close to that of the BiCausalTree in most cases. It is possible to apply post hoc models for explaining non-interpretable methods such as IPW. However, this requires using models on top of models. One method to get such explanations is using the SHAP values [1], but other methods also exist [2]. We note, however, that these methods provide explanations and not an inherent interpretation of the model. Using explanation models, it is possible for the most explainable feature to end up having no actual effect on the model decision.
>
>
> references:
>
> [1] Syrgkanis, Vasilis, et al. "Causal inference and machine learning in practice with econml and causalml: Industrial use cases at microsoft, tripadvisor, uber." Proceedings of the 27th ACM SIGKDD conference on knowledge discovery & data mining. 2021.
>
> [2] Bodria, F., Giannotti, F., Guidotti, R., Naretto, F., Pedreschi, D., & Rinzivillo, S. (2023). Benchmarking and survey of explanation methods for black box models. Data Mining and Knowledge Discovery, 1-60.

---

> > ### Comment · Reviewer_aknA · 2023-08-18
> > **thanks for the reply**
> >
> > The authors addressed all my concerns and I would like to keep my positive score.

---

### Official Review · Reviewer_kdjD · 2023-07-07

**Soundness:** 3 good
**Presentation:** 3 good
**Contribution:** 3 good
**Rating:** 6
**Confidence:** 3

**Summary:**

This paper focuses on achieving interpretable causal effect estimation, where the goal is to ensure that each decision within the algorithm is explicit and traceable. The authors propose a decision tree-based balancing method to address this problem, which identifies clusters where local natural experiments occur. The effectiveness of the proposed algorithm is empirically evaluated using synthetic and semi-synthetic data. The paper also did several ablation studies on the trade-off between interpretability and bias and the consistency of the decision tree.

**Strengths:**

Overall, this paper is well-executed and demonstrates several notable strengths.
- This paper is very clear. It effectively presents complex ideas in an easily understandable manner.
- The problem addressed in the paper is well-motivated and interesting.
- The authors display a strong grasp of the related work in the field, effectively positioning their contribution within the existing literature.
- The empirical analysis conducted in the paper is thorough and yields valuable insights.
- The method is intuitive and well explained.

**Weaknesses:**

- Style file: One issue with this paper is that it doesn't follow the NeurIPS style guidelines, specifically regarding paragraph spacing. The paragraphs are not well-separated, which makes it harder to read and understand the content. This affects the overall flow and coherence of the paper. Additionally, the excessive content allowed due to the spacing issue may be seen as unfair to authors who followed the style guide correctly. This may be a potential ground for rejection.


- Method: The rationale behind considering features with the highest ASMD as potential confounders is not well-explained. This is an important assumption in the paper, but it lacks a clear justification or empirical investigation. Providing additional explanations or conducting empirical studies would strengthen this aspect of the paper.

- The experiment section of the paper is not self-contained. Although the motivating problem revolves around identifying subpopulations with natural experiments, this aspect is not adequately illustrated in the experiment section. Instead, the focus is primarily on bias analysis, with the analysis related to interpreting the causal effect estimation process deferred to the appendix. This undermines the fulfillment of the paper's fundamental promise to the readers. To address this issue, it is highly recommended that the authors integrate the analysis into the main paper, ensuring that the key components align with the paper's core premise.


**Questions:**

Why does a feature with high ASMD more likely to be a confounder?

---

> ### Author Rebuttal · Authors · 2023-08-08
>
> We thank the reviewer for their helpful comments and suggestions, and for time spent reading our work.
>
> - paragraph spacing:
> We thank the reviewer for noticing, we have found the source for this error and fixed the spacing, leaving the main text untouched (by slightly resizing the figures).
>
> - Using the highest ASMD feature as the splitting feature:
> ASMD is a well established measure of balance between treatment groups in the field ([1], [2], [3]). The reason we choose the feature with the highest ASMD is that we assume (like most other methods in causal inference) that the measured covariates are indeed confounders. Therefore, the covariate with the highest ASMD is most likely to cause the most confounding bias, and therefore adjusting for it will likely minimize the residual confounding bias the most. We have added a supplementary figure showing this in practice, where we use a random feature split instead of the feature with the highest ASMD, and show that both in terms of estimation bias and in terms of feature imbalance, using the highest ASMD feature as the splitting feature gives better performance (see plot in additional plots pdf of the rebuttal).
> In addition, we will explain the motivation better and clarify that in the text, since apparently it is currently not sufficiently explained (line 164, please see point 4 in the general rebuttal).
>
> - Focus is on bias analysis instead of identifying subpopulations with natural experiments:
> As correctly pointed out by the reviewer, the main motivation of the method is identifying subpopulations with natural experiments. In such subpopulations, it is expected that the confounding bias between treated and control is minimized, leading to an unbiased estimate of the effect. Thus, in order to check whether the model was indeed able to identify natural experiments, a good proxy is its bias from ground truth effect (please also see point 2 in the general rebuttal). Indeed, we demonstrate in a synthetic dataset, where the natural experiment subpopulations are clearly defined, the ability of the tree to discover them (see figure 1a), and show how this can occur also on real-world datasets where we do not have the ground truth of the subpopulations that constitute natural experiments, by proxy of an unbiased effect estimation. We will revise the text to make this connection clearer.
>
> - Interpretation of the causal effect estimation process deferred to the appendix:
> We agree with the reviewer that a critical strength of the model is its interpretability, and is one of the motivations to develop it, and that it would be beneficiary to demonstrate that in the main text. However, due to space limitations, we could not add the tree structures and their explanations in the main text. We would like to emphasize that the ability to critique and evaluate the decisions made by the model, by experts and decision makers (e.g., criticizing the subpopulations created by it and which ones we can infer on and the ones we cannot due to positivity violations) is a unique feature of this model that is imperative in sensitive domains. In the absence of ground truth, as is the case in real world data applications, this is highly desired and important. We will add further discussion on this in the text as this is indeed a pivotal point of our model.
>
>
> references:
>
> [1] Austin, P. C. “Balance diagnostics for comparing the distribution of baseline covariates between treatment groups in propensity-score matched samples,” Statistics in medicine, vol. 28, no. 25,483pp. 3083–3107, 2009.
>
> [2] Austin, P. C. (2009). Using the standardized difference to compare the prevalence of a binary variable between two groups in observational research. Communications in statistics-simulation and computation, 38(6), 1228-1234.
>
> [3] Ali, M. S., Groenwold, R. H., Belitser, S. V., Pestman, W. R., Hoes, A. W., Roes, K. C., ... & Klungel, O. H. (2015). Reporting of covariate selection and balance assessment in propensity score analysis is suboptimal: a systematic review. Journal of clinical epidemiology, 68(2), 122-131.

---

> > ### Comment · Reviewer_kdjD · 2023-08-15
> > **Thank you for the response**
> >
> > ``the covariate with the highest ASMD is most likely to cause the most confounding bias, and therefore adjusting for it will likely minimize the residual confounding bias the most.`` This claim is not obviously true to me. Is there a proof, empirical studies, or existing work that demonstrates that?

---

> > > ### Author Response · Authors · 2023-08-16
> > >
> > > We are not aware of existing work that demonstrates this, but it is well accepted in the field, and we believe this behavior can be easily simulated.
> > > The following code snippet shows that adjusting for the confounder with the larger imbalance (larger ASMD) results in lower estimation bias of the treatment effect, i.e., less confounding bias (since confounding is the only source of bias in this case). In contrast, adjusting for the confounder with the smaller ASMD results in far more biased treatment effect estimation, i.e., the residual confounding bias left is larger.
> > >
> > > ```python
> > > import numpy as np
> > > import pandas as pd
> > > import statsmodels.api as sm
> > >
> > > def generate_data(N=1000, seed=0):
> > >     rng = np.random.default_rng(seed)
> > >
> > >     a = rng.binomial(1, 0.5, size=N)
> > >     X = rng.normal(0, 1, size=(N, 2))
> > >     X[a==1, 1] += 2  # x1 has larger discrepancy between treated and control units
> > >     y = X[:, 0] + X[:, 1] + a
> > >
> > >     X = pd.DataFrame(X, columns=["x_smallASMD", "x_bigASMD"])
> > >     a = pd.Series(a, name="a")
> > >     y = pd.Series(y, name="y")
> > >     return X, a, y
> > >
> > > def calculate_asmd(X, a):
> > >     is_treated = a == 1
> > >     X1 = X.loc[is_treated]
> > >     X0 = X.loc[~is_treated]
> > >     smds = (X0.mean() - X1.mean()) / np.sqrt(X0.var() + X1.var())
> > >     asmds = smds.abs()
> > >     return asmds
> > >
> > >
> > > X, a, y = generate_data()
> > > data = X.join(a).join(y)
> > >
> > > calculate_asmd(X, a)
> > > >>> x_smallASMD    0.031651
> > >     x_bigASMD      1.386414
> > >
> > > # Adjusting for both covariates retrieves the true effect:
> > > print(sm.formula.ols("y ~ x_smallASMD + x_bigASMD + a", data=data).fit().params["a"]) # 1.00
> > > # Adjusting for the more-biased covariate leads to a result somewhat closer to the true effect:
> > > print(sm.formula.ols("y ~ x_bigASMD + a", data=data).fit().params["a"])  # 0.92
> > > # Adjusting for the less-biased covariate leads to high estimation bias
> > > print(sm.formula.ols("y ~ x_bigASMD + a", data=data).fit().params["a"])  # 2.98
> > > ```
> > >
> > > Additionally, this comment touches on an important point, further suggested by the above demonstration.
> > > Future work can combine the covariate-outcome associations with the ASMD in order to select the best candidate for splitting. For example, multiplying the ASMD of covariate _j_ with the absolute regression coefficient of (a standardized) covariate _j_, and selecting the covariate maximizing this combined value.
> > > we considered this approach, but decided against it in order to obtain an outcome-agnostic method.

---

### Official Review · Reviewer_vmTU · 2023-07-07

**Soundness:** 1 poor
**Presentation:** 2 fair
**Contribution:** 2 fair
**Rating:** 3
**Confidence:** 3

**Summary:**

The paper introduces a decision tree methodology to identify regions where selection bias no longer ensures covariate balance. These regions, which have some level of interpretability, can then be removed in subsequent analysis.

**Strengths:**

The paper presents an interesting decision tree methodology.

The paper contains a significant amount of simulation experiments to validate the procedure.

**Weaknesses:**

Despite a focus on covariate balance, there is no guarantee of balance unlike competing methods (rerandomization, matching, etc). Furthermore, there is limited analysis to show the claims of balance are fulfilled, especially for high dimensions.

The paper explores a bias-interpretability tradeoff, but provides no rigorous definition. The proposed model often is more biased than alternative models, most notably IPW, and it's not clear that the resulting decision trees, or their interpretations, are actually sensible. No discussion of estimator variance is given or how that might factor into a tradeoff, despite the high variance generally expected from decision tree estimators.


**Questions:**

In Section 2, it is briefly claimed that BICauseTree is better suite for ATE estimation instead of CATE, with no further discussion. This feels like a potentially important point and requires further justification.

Trimming has the potential to bias treatment effect measurement. Are there any guarantees that trimming in your model ensures unbiased estimates?

Are the decision trees in Figures A13 and A17 interpretable? While the decision tree can be followed, the trimmed regions seem to have limited clinical interpretation. More discussion around this feels necessary. In particular, interpretable does not seem to imply correctness for this method.

ASMD is often minimized to establish balance. While splitting on variables that exhibit high ASMD should generally lead to some level of balance, is there any guarantee that the resulting regions will be optimally balanced in any sense?

**Limitations:**

The paper touches on some limitations, most notably the increases bias that is expected. The paper does not discuss the variance of the estimator in detail relative to other methods, which is another potential limitation.

The method advocates for trimming nodes that violate positivity. These nodes could contain sensitive subpopulations and could lead to fairness concerns.

---

> ### Author Rebuttal · Authors · 2023-08-08
>
> We thank the reviewer for their time, helpful comments and insights.
>
> - Guarantee of balance:
> As pointed, our method cannot guarantee covariate balance. This is true for most practical methods; Matching, for instance, guarantees balance only if done by exact matching on the covariates, which is impractical in high dimensions and can result in a low remaining sample size. In practice, matching is more often done using some distance function (KNN, propensity score matching, etc.) where no guarantee of balance can be given.
>
> - Analysis of balance:
> We show some analysis of balance in appendix Figures A6, A12, A16, A21. Also, we argue that balancing is not the goal but a tool, while the real focus is unbiased estimation.
>
> - Balance in high dimension:
> As mentioned, we do not claim that the tree will result in balanced sub-populations in all cases, but in most cases where some natural experiments exist in the data. This is explicitly shown for low-dimensional synthetic data, but implicitly assumed by the performance on other datasets with more covariates. To demonstrate this point with more evidence, we added a supplementary figure showing the performance of BICause Tree in high dimensions, where as we increase the dimension, the tree maintains its performance both in terms of balancing and debiasing, when natural experiments exist (see plots in plots pdf of the rebuttal). The reason this is true is that the tree effectively performs on a sub-space of the features that are un-balanced, and not on the whole feature space.
>
> - Definition of bias-interpretability tradeoff:
> We discuss the trade off between interpretability and bias in lines 297-299 and show it in figure 4- the complexity of the tree compared to its bias. A similar trade-off exists for more complex models: they can achieve better performance in estimation bias but are less interpretable than our model. Our model’s interpretability is traded off against some estimation bias, due to its non-parametric nature. We agree that further discussion on this point would be useful and we'll add it to the paper.
>
> - Sensibility of BICause tree interpretation:
> We argue that the ability to tell if the estimation of a causal model is sensible is an inherent  problem in causal inference due to the lack of ground truth. Our method, however, allows inspecting whether the decision is sensible or not by experts, because it is interpretable and transparent, whereas other methods that are not interpretable cannot be inspected as such. This is a key motivation for using our model instead of others- in the absence of ground truth, expert inspection of the algorithm decisions is the next best option, and this is what we allow.
>
> - Discussion of estimator variance:
> We note that estimator’s variance is shown relative to other estimators (Figures 1, 2, 3), and discussed (lines 272, 276, 295-6).
> As we are using a tree-based model, we indeed inherit all of the advantages as well as disadvantages of trees, e.g. high sample size required, and higher variance.Specifically, the trees’ variance is comparable with IPW, which is also known as high-variance estimator [1].
> Regarding high dimensional data, our model should encounter less difficulties than other estimators as the tree ultimately picks a small subset of features to separate the population, if natural experiments exists. We added a supplementary figure showing this, namely that the performance of the model remain unaffected when increasing dimension (see plots pdf of the rebuttal).
>
> - BICauseTree is better suited for ATE estimation than CATE:
> We discuss CATE in the text to emphasize the difference from Causal Trees which are another custom-objective function tree-based model, that optimizes for effect heterogeneity, whilst our optimizes for exchangeability (lines 119-128).
> In fact, we made a mistake in the sentence phrasing and will clarify it (line 124) - BICause Tree might not necessarily be better suited for ATE than Causal Trees, but can only estimate ATE. However, ATE is often an estimand of interest, thus we believe our model is useful.
>
> - Effect of trimming on effect measurement:
> Trimming is a common method for positivity identification in causal inference. Instead of performing this step before applying the causal model (as one would do in IPW, for instance), it is performed as an inherent step in our model. We argue that this is imperative, since we cannot make data-based inferences on units without overlap. We stress that all comparisons to other models in the paper are done on the basis of the same trimmed population as in the BICause tree, to ensure a fair comparison (lines 207-210).
>
> - Interpretability of trees (Figures A13 and A17) and their correctness:
> Ideally, the model can give interpretable regions. This of course depends on the data. The trees in A13 and A17 are indeed interpretable and this allows experts to criticize/evaluate the correctness of the method, which cannot be done in methods that are not interpretable by design. In general, interpretable does not imply correctness in any model. It does allow criticizing the decisions made by the model, which is impossible when using other, non-interpretable models.
>
> - Guarantee that the resulting regions will be optimally balanced:
> Unfortunately, we could not establish an analytical guarantee (note that most practical methods do not have such guarantees). We do demonstrate that when natural experiments exist in the data, the model can find them and lead to good balancing.
>
> - Fairness concerns due to the trimmed nodes:
> As explained above, trimming is a mandatory step in causal inference as it is impossible to make valid causal claims when there is no overlap. It is often performed anyway using opaque models. Our advantage over other models is the capability to precisely define, in the covariate space, those populations we can or cannot infer on.
>
> [1] Khan et al. "Adaptive normalization for IPW estimation" Journal of Causal Inference (2023)

---

> > ### Comment · Reviewer_vmTU · 2023-08-21
> >
> > Thank you for your detailed reply. Most of my concerns have been addressed to some degree, although I still have significant concerns:
> > - I acknowledge that balance is a secondary concern to the primary goal of unbiassdness. That said, BICauseTree is explicitly referred to as an "interpretable balancing method". The lack of any type of guarantee, for balance or unbiassdness in general, along with unconvincing empirical results, remains a significant concern for me.
> > - The lack of a rigorous definition of interpretability, or the bias-interpretability tradeoff, makes any claims of interpretability difficult to evaluate. Our difference of opinion in the interpretability of Figures A13 and A17 further reinforces this.
> > - Furthermore, I still do not fully understand how the improved interpretability of excluded observations prior to ATE estimation will significantly aid a practitioner. Is it even an ATE estimator at that point? Is your focus on situations where a CATE estimator is impractical? I believe further discussion of the practical implications of BICauseTree is necessary.
> >
> > Ultimately, after reviewing your response, and the feedback of my fellow reviewers, I will not be changing my initial score at this time.

---

> > > ### Author Response · Authors · 2023-08-21
> > >
> > > Thank you for your additional consideration and the points brought up. Please see our point-by-point response:
> > >
> > > * We understand the reviewer's concern about the lack of analytical guarantees and are only left with restating the fact that neither do other well-known modern methods [1 (a NeurIPS paper too), 2] that show their utility using simulations alone. If the reviewer can suggest specific experiments they would like to see to further convince them, then we will be happy to conduct them.
> > >
> > > * We operate within the page-limit constraints, and we acknowledge we might have been terse on some definitions we thought to be either well-established, intuitive, or can be deferred to external resources. For example, we describe interpretability based on Cynthia Rudin's landmark paper [3] and further refer to it. In her paper, she claims that _"Interpretability is a domain-specific notion so there cannot be an all-purpose definition"_ and that it is basically a useful constraint that _"obeys structural knowledge of the domain"_.
> > > We further believe that the fact we can discuss whether the _content_ of the explanation makes sense is a big step forward as this discussion is not even possible to begin with under any other black-box model.
> > >
> > > * The positivity assumption is one of the three assumptions required in order to make valid data-driven causal claims (in addition to causal consistency and exchangeability). However, in real-world data, not all units are always comparable to begin with, as they may have no counterparts in the other group to allow extrapolation of the outcome across groups. It is a very common in practice to discard these units [4, 5]. However this indeed changes the actual eligibility criteria and therefore to whom we believe the results will generalize to in the population. This is the reason why it is of interest to know on whom exactly the causal claims are made. The overlapping region is where a researcher believes their results will transport from the sample to the population (i.e., for whom the results are externally valid) [6]. When we discuss transportability, the "ATE" is indeed not well-defined, and need to be separated to the sample ATE (SATE) and the population ATE (PATE) [7, 8].
> > > Lastly, the CATE is not a concern in this study. We again stress the ATE is a valid estimand on its own. For example, it has both logistical and philosophical justification in the field of public health policy.
> > >
> > >
> > > [1] Shi, Claudia, David Blei, and Victor Veitch. "Adapting neural networks for the estimation of treatment effects." Advances in neural information processing systems 32 (2019).
> > >
> > > [2] Hill, Jennifer L. "Bayesian nonparametric modeling for causal inference." Journal of Computational and Graphical Statistics 20.1 (2011): 217-240.
> > >
> > > [3] Rudin, Cynthia. "Stop explaining black box machine learning models for high stakes decisions and use interpretable models instead." Nature machine intelligence 1.5 (2019): 206-215.
> > >
> > > [4] Potter, Frank J. "The effect of weight trimming on nonlinear survey estimates." Proceedings of the American Statistical Association, Section on Survey Research Methods. Vol. 758763. Washington, DC: American Statistical Association, 1993.
> > >
> > > [5] Cole, Stephen R., and Miguel A. Hernán. "Constructing inverse probability weights for marginal structural models." American journal of epidemiology 168.6 (2008): 656-664.
> > >
> > > [6] Oberst, Michael, et al. "Characterization of overlap in observational studies." International Conference on Artificial Intelligence and Statistics. PMLR, 2020.
> > >
> > > [7] Degtiar, Irina, and Sherri Rose. "A review of generalizability and transportability." Annual Review of Statistics and Its Application 10 (2023): 501-524.
> > >
> > > [8] Imai, Kosuke, Gary King, and Elizabeth A. Stuart. "Misunderstandings between experimentalists and observationalists about causal inference." Journal of the Royal Statistical Society Series A: Statistics in Society 171.2 (2008): 481-502.

---

### Author Rebuttal · Authors · 2023-08-08

We thank the reviewers for their time, helpful comments and for sharing their expertise.

We present in our work a model for effect estimation in observational data that is inherently interpretable, scalable, and has the useful consequence of abstaining from inference on subgroups where inference would be unreliable (i.e., with positivity violations).
Here, we address the points raised by multiple reviewers and present the changes made following these comments:

1. Limitations and strengths of decision trees: Our method builds on top of decision trees for their inherent interpretability. However, as noted by the reviewers, by doing so we also inherit their disadvantages. Namely, their requirement for a larger sample size to estimate densities non-parametrically (and therefore their higher variance relative to parametric models), and their inappropriateness for certain functional forms (e.g., estimating straight lines inefficiently). Nevertheless, these limitations can be partially overcome by fitting estimators at leaf nodes (similar to how one may convert a piecewise constant function to a piecewise linear one).
We note that with the limitations we also inherit the strengths of trees: their interpretability, their ability to deal with high dimensional data by looking at a relevant sub-space of the features, and their ability to account for complex interactions between features.

2. Several reviewers questioned why we focus on reduction of estimation bias rather than improvement in balancing. We note that although our method is guided by improvement in balancing, our ultimate goal (like any causal inference method) is the consequent reduction in estimation bias. Therefore, estimation bias is the focus metric of our results as it provides stronger evidence for the usefulness of our method. We did, of course, evaluate balancing as well, but due to space limitations, this part is currently in the appendix.

3. Two of the reviewers commented on the correctness, or sensibility, of our model. The interpretability of our model was a key motivation for developing it and we believe that it is a key strength of it. Important to note, however, that interpretability does not imply sensibility or correctness. It does, however, provide the possibility for decision makers and domain experts to criticize the inferences made by the model and their correctness - something that is not available with black-box models even with explainability methods. This is even more important in causal inference, where there is no inherent ground truth to validate the results of the model. Thus, in summary, our model allows subjecting the algorithm decisions to such inspection, whereas other, non-interpretable models, do not offer this important capability.

4. Some of the reviewers questioned why we focus on the largest ASMD. The idea behind our method is to gradually reduce bias by stratifying on the most imbalanced feature. The intuition is that the most imbalanced confounder causes the most confounding bias. ASMD is a common method to evaluate imbalance in applied causal inference, but our motivation above was not well phrased in the manuscript, leading to some unclarity regarding our choice to use it. We have made clarifications in the text to explain the motivation for using ASMD. Furthermore, to support our algorithmic choice, we implemented a variant of our method that chooses features to split on randomly, rather than taking the one with highest ASMD, and provide the figures to show that it underperforms both in terms of estimation bias and overall balance. This analysis will be added to the appendix. (see last item regarding new experiments for more details).

5. The rebuttal pdf includes two new results. One, described above, shows that selecting to split on the feature with highest ASMD at each recursion step leads to better results both in terms of estimation bias and in terms of balancing, and is therefore justified. The second result strengthens our claim of scalability. We claim our methods is suited for high-dimensional data, but only present somewhat implicit evidence. We now show that our method rediscovers existing natural experiments even when increasing the dimensionality of the data by two orders of magnitude.

---

### Decision · Program_Chairs · 2023-09-21

**Decision:**

Reject

**Comment:**

The paper studies how to reduce treatment allocation bias using decision trees. While the question is of potential interest, the paper demonstrates limited methodological novelty, without much theoretical justification or convincing empirical results. In particular, while the paper argues that their method is interpretable and can handle overlap issues, but neither point is demonstrated convincingly in empirical studies (e.g. synthetic data experiments with varying levels of overlap).